# Exposure of *Pseudomonas aeruginosa* to Cinnamaldehyde Selects Multidrug Resistant Mutants

**DOI:** 10.3390/antibiotics11121790

**Published:** 2022-12-10

**Authors:** Alexandre Tetard, Susie Gaillot, Eline Dubois, Soumaya Aarras, Benoît Valot, Gilles Phan, Patrick Plésiat, Catherine Llanes

**Affiliations:** 1UMR CNRS 6249 Chrono-Environnement, Agents Pathogènes, Faculté de Médecine-Pharmacie, Université Bourgogne Franche-Comté, 25000 Besançon, France; 2Plateforme de Bioinformatique et Big Data au Service de la Santé, Faculté de Médecine-Pharmacie, Université de Bourgogne Franche-Comté, 25000 Besançon, France; 3UMR CNRS 8038, CiTCoM, Université de Paris, 75006 Paris, France

**Keywords:** *Pseudomonas aeruginosa*, essential oils, cinnamaldehyde, antibiotic resistance, efflux

## Abstract

Cinnamaldehyde (CNA), the main component of cinnamon essential oil, is one of the most active plant compounds against nosocomial pathogen *Pseudomonas aeruginosa*. Exposure of wild-type strain PA14 (MIC 700 µg/mL) for 5 to 10 days to fixed (900 µg/mL) or increasing (from 900 to 1400 µg/mL) concentrations of this natural antibacterial resulted in emergence of resistant mutants CNA-A1 to A3, and CNA-B1 to B7, respectively. Genome sequencing experiments showed that each of CNA-A1 to A3 mutants differed from PA14 by one SNP, and a slight increase in CNA resistance level (from 700 to 900 µg/mL). By comparison, mutants B1 to B7 were more resistant (up to 1100 µg/mL); each of them harbored multiple SNPs (from 24 to 39) likely as a consequence of alteration of DNA mismatch repair gene *mutS*. Of the ten mutants selected, eight contained mutations in gene *nalC*, which indirectly downregulates expression of the operon that codes for multidrug efflux system MexAB-OprM, and showed increased resistance (up to 16-fold versus PA14) to antibiotic molecules exported by the pump, including ß-lactams and fluoroquinolones. Of the six mutants with the highest CNA resistance, five were no longer motile because of alteration of genes *flgJ, fliE* and/or *pilJ* genes. Altogether, our data show that *P. aeruginosa* is able to adapt to strong electrophilic molecules such as CNA by upregulating its intrinsic efflux pump MexAB-OprM, and through less well-characterized pleiotropic changes. Whether multidrug-resistant mutants can emerge in patients using cinnamon essential oil as self-medication needs to be assessed further.

## 1. Introduction

Efficacy of anti-infectious treatments tends to be compromised by the sustained emergence and subsequent diffusion of bacterial strains resistant to multiple antibiotics (World Health Organization, 2018, www.who.int). To address the global Public Health issue of multidrug resistance, several strategies have been envisaged including the reappraisal of a number of natural drugs used in traditional medicine [1]. In this context, several aromatic plants extracts (namely, essential oils, EOs) were found to have interesting in vitro antibacterial properties when used alone [2], mixed together [3], or combined with antibiotics [4,5]. The composition of EOs varies according to complex factors linked to the genetics of plants themselves, where and how they grow. The biological activities of EOs mostly reflect those of their main components such as cinnamaldehyde and cinnamic acid (in cinnamon oil); carvacrol, thymol, γ-terpinene and ρ-cymene (oregano and thyme oils); eugenol (clove oil); terpinen-4-ol, γ-terpinene, α-terpinene and 1,8-cineole (tea tree oil); calamenene and leptospermone (manuka oil) [5].

The mode of action and composition of EOs have raised increasing interest over the last years. EO exposure often results in profound damage to the bacterial cell wall, alteration of membrane proteins, leakage of the cellular content, dissipation of the motive proton force, and coagulation of the cytosol [6]. Specific physiological damages may also occur, such as the inhibition of efflux pumps with geraniol [7], reduced production of exoproteins with eugenol in *Staphylococcus aureus* [8] and inhibition of amino acid decarboxylases with cinnamaldehyde [9]. In general, higher EO concentrations are necessary to inhibit the growth of, or to kill Gram-negative bacteria as compared with Gram-positives [10], suggesting a protective role for the bacterial outer membrane (i.e., an additional lipid bilayer not present in Gram-positives).

The Gram-negative opportunistic pathogen *Pseudomonas aeruginosa* is notably resistant to many EOs that are otherwise active against Enterobacterales and Gram-positive species [11]. In this environmental bacterium, the poor outer membrane permeability acts in synergy with several RND (Resistance Nodulation cell Division) active efflux systems, including the major pump MexAB-OprM, to restrict the penetration of xenobiotics [12,13,14]. Whether higher levels of resistance may be gained by *P. aeruginosa* when exposed to EOs remains unclear. Furthermore, the issue of mutagenic properties of EOs (i.e., their capacity to increase the mutation rates of microorganisms) needs to be clarified as well. Previous experiments, in which strains of *P. aeruginosa* were exposed to toxic concentrations (up to 10 mg/mL) of oregano or cinnamon EO failed to isolate more resistant mutants [15,16]. In contrast to antibiotics, EOs exert pleiotropic effects on bacterial cells, which may explain why selection of such mutants is difficult.

Because of its high bactericidal activity, cinnamon oil has been proposed as a possible alternative to antibiotics for fighting mild *P. aeruginosa* infections (MIC from 400 to 1000 µg/mL) [15,17]. However, in the present study we demonstrate that *P. aeruginosa* is able to withstand lethal CNA concentrations through a selection of mutations, some of which increase expression of the multidrug efflux system MexAB-OprM. Although the gain in CNA resistance remains modest, it may be associated with decreased activity of several antibiotics.

## 2. Results

### 2.1. In Vitro Selection of Mutants Showing an Increase Resistance to CNA

Because of their pleiotropic action, we first wondered whether electrophilic compounds such as CNA might have potential mutagenic properties on *P. aeruginosa*. To test this hypothesis, reference strain PA14 (MIC of CNA = 700 µg/mL) was cultivated in presence of subinhibitory (500 µg/mL) concentrations of CNA, and streptomycin-resistant mutants were counted on selective plates. Mutant rates were similar with (4.9 × 10^−10^, *p* < 0.05) or without CNA (2.3 × 10^−10^), thereby indicating that this agent is not mutagenic to *P. aeruginosa.*

Exposure of liquid cultures of PA14 to 1.3-fold the MIC of CNA (i.e., 900 µg/mL) for 24 or 48 h did not allow the selection of CNA resistant mutants. This goal was achieved after a ten-day incubation, and by diluting the bacterial cultures with fresh MHB medium containing 900 µg/mL CNA (method A). Three clones were retained for further analysis (CNA-A1, -A2, -A3). In parallel, serial passages of strain PA14 in MHB containing increasing CNA concentrations (method B) led to the isolation of three resistant B mutants (CNA-B1, -B2 and B3) after five days at a concentration of 1100 µg/mL, and four additional B mutants (CNA-B4, -B5, -B6, -B7) after ten days (at 1400 µg/mL).

The ten mutants from methods A and B displayed only a marginal increase in CNA resistance (from 1.1- to 1.6-fold that of PA14) (Table 1), thus highlighting the difficulty for *P. aeruginosa* to set up additional defense mechanisms to those involved in its natural resistance toward the agent.

### 2.2. CNA Can Select Multidrug Resistance Mutants

Interestingly, the CNA-resistant mutants reported above appeared to differ from their parent PA14 by their susceptibility to various antibiotics. Most (8/10) were less susceptible (from 4- to 16-fold) to the β-lactam ticarcillin and to the fluoroquinolone ciprofloxacin (Table 1). In addition, some B mutants were more resistant to imipenem and colistin (from 2- to 4-fold), while 6/10 mutants were more susceptible (2- to 4-fold) to the aminoglycoside gentamicin.

To gain insight into the role of RND efflux pumps in these phenotypic changes, we determined the expression levels of genes encoding components of four efflux pumps known to export antibiotics in *P. aeruginosa*, and to confer multidrug resistance when overproduced: *mexB* for MexAB-OprM (substrates: β-lactams except imipenem, and fluoroquinolones), *mexY* for MexXY/OprM (cefepime, aminoglycosides and fluoroquinolones), *mexE* for MexEF-OprN (fluoroquinolones), and *mexC* for MexCD-OprJ (cefepime and fluoroquinolones). Accounting for their higher resistance to ticarcillin and ciprofloxacin, *mexB* appeared to be significantly overexpressed in the eight mutants exhibiting this phenotype. Additionally, the low expression of *mexY* (from 2- to 10-fold less than in PA14) in six mutants was correlated to their hypersusceptibility to aminoglycosides (Table 1).

### 2.3. Genetic Traits of CNA Resistant Mutants

The three mutants from method A were found to harbor single but different SNPs with respect to PA14, while the seven method B mutants diverged from PA14 by 16 to 39 SNPs, likely as a result of alteration of DNA mismatch repair gene *mutS* (Table 2). A Q_589_R substitution predicted to have a slight impact on the MutS protein (−2.456 neutral PROVEAN score) was detected in CNA-B2, while the six other B mutants contained a same *mutS* allele lacking 9 nucleotides (∆_2327–2336_). Loss of function of this protein (as well as MutL and UvrD) is known to increase mutations rates in *P. aeruginosa*, especially in long-term chronic infections [18]. Supporting the notion that inactivation of MutS boosted accumulation of multiple mutations in B clones, 16 to 26 SNPs were identified in bacteria after five days of CNA exposure (CNA-B1 to -B3), and 24 to 39 SNPs after ten days (CNA-B4 to -B7). A phylogenetic tree showing the relationships of these mutants is shown in Appendix A. With the exception of CNA-B2, all the B mutants were closely related (e.g., CNA-B3/B5 and CNA-B4/B6).

### 2.4. Role of MutS/L in Emergence of CNA-Resistant Mutants

To confirm the role of hypermutability in emergence of CNA resistance, we carried out additional in vitro evolution experiments with two DNA mismatch repair defective constructs of PA14 (PA14*mutS::*MAR2xT7 and PA14*mutL::*MAR2xT7), exposed to a fixed 900 µg/mL CNA concentration [13]. CNA-resistant clones were obtained at rates of 1.0 × 10^−7^ and 8.6 × 10^−7^, respectively, while no resistant colonies developed from parental strain PA14 (<1.0 × 10^−9^). Hypermutator mutants are frequent in chronic infections such as those affecting the airways of cystic fibrosis (CF) patients [19]. To gain insight into whether CNA resistance could emerge in this context, we repeated the same experiments with three clonally related *P. aeruginosa* isolates collected longitudinally in a same CF patient at the early (P6-1), mid (P6-14) and late (P6-21) phases of lung colonization. As expected, mutant rates were much higher in the MutS deficient (L_255_Q) isolates P6-14 and P6-21 (3.2 × 10^−7^ and >5.4 × 10^−5^, respectively) than in the early isolate harboring wild-type *mutS* and *mutL* alleles (<1.0 × 10^−9^). Susceptibility testing of 68 CNA-tolerant clones by the disk diffusion method (six from P6-1, and 62 from P6-21) revealed a high diversity of phenotypic profiles due to increased or decreased resistance to one or more antibiotic molecules of a same class or of different classes (Appendix A). Analysis of the mechanisms behind these multiple phenotypes is ongoing. However, as such, our results strongly suggest that CNA may have profound effects on the resistance profiles of hypermutator subpopulations found in the CF lung.

### 2.5. Role of NalC in Bacterial Adaptation to CAN

Eight out of ten mutants from the A and B selection protocols appeared to possess mutations in *nalC*, a gene which indirectly represses operon *mexAB-oprM* expression. The *nalC* product negatively controls transcription of a gene called *armR* that codes for a small protein (ArmR) able to sequester the local repressor of *mexAB-oprM*, MexR [20]. Confirming the activation of the NalC regulatory pathway in our mutants, genes *mexB* and *armR* were upregulated (from 2.5 to 7.6-fold, and from 24 to 423-fold, respectively) as compared with parental strain PA14. The NalC-dependent overexpression of *mexAB-oprM* thus accounts for the cross-resistance of most CNA-selected clones to ticarcillin and ciprofloxacin, two substrates of MexAB-OprM (Table 1). Rather surprisingly, three different mutations targeted gene *nalC* in evolution experiments A and B, while other regulatory genes of operon *mexAB-oprM*, such as *mexR* and *nalD*, remained unaffected. In 7/8 *nalC* mutants, the ORF was disrupted by a frameshift (+C_450_ or −A_486_). The remaining mutant CNA-A2 exhibited an amino acid substitution (T_24_P) in the gene product. This amino acid residue is not conserved in the TetR family of transcriptional regulators, to which NalC belongs (Appendix A), but it is situated in a helix (Hα1) of the DNA-binding domain of TetR repressors [21]. To better evaluate the impact of the T24P substitution on NalC oligomerization or DNA-binding, we mapped this mutation on a three-dimensional dimeric model bound to a cognate DNA. As the structure of NalC from *P. aeruginosa* has not been determined yet, we used the dimeric crystal structure of a homologous TetR protein from *Mesorhizobium japonicum* (PDB code 3BHQ, 24.4% of sequence identity), that also has a threonine at position 24 (Appendix A). According to our structural model, T_24_ is not expected to be involved in dimeric interface, or to be in direct contact with DNA. Nevertheless, its substitution by a proline residue could destabilize the DNA-binding domain (located right next door, residues 27–48 overlapping Hα1, Hα2 and Hα3) by introducing a structural kink in helix α1, because of the hydrogen bond loss of the main chain. Thus, the substitution T_24_P might trigger the disruption of NalC-DNA binding. In addition, T_24_ could interact through a hydrogen bond with the residue R_106_ of helix α7, which is part of the ligand-binding domain (Appendix A). This time, the substitution T_24_P could indirectly disturb the binding pocket of NalC by disconnecting the hypothetical interaction with R_106_.

Despite preferential selection of *nalC* mutants over *nalB* or *nalD* clones in presence of CNA, activation of the NalC pathway does not seem to specifically confer higher resistance levels to the biocide. As indicated in Table 1, CNA MICs were indeed very similar between the *nalB* mutant PA14*mexAB^+^* (900 µg/mL) and our eight *nalC* clones (ranging from 900 to 1100 µg/mL), which might indicate that the NalC regulatory circuit provides *P. aeruginosa* with other physiological advantages for its adaption to CNA. In search of other alterations accounting for the slightly higher CNA resistance of mutants CNA-B1 and CNA-B3 to -B7 (1000–1100 µg/mL), we deleted the two genes *mexAB* in frame in all of these clones. The third ORF of the operon, *oprM*, was left intact because its product interacts with several other RND transporter/periplasmic adaptor pairs to form functional tripartite efflux systems [22]. Interestingly, in contrast to parental strain PA14 (mutant PA14∆AB), inactivation of MexAB-OprM had a lesser impact on the growth of our eight mutants on 500 µg/mL CNA (Figure 1), an observation that pleads for the contribution of additional CNA resistance mechanisms in the *nalC* clones.

Since *P. aeruginosa* is able to degrade CNA into its less toxic metabolite cinnamic alcohol [13], we wondered whether some CNA-selected mutants might be more active in this detoxification process. However, thin layer chromatography analysis of culture supernatants did not show a clear relationship between the levels of CNA resistance and extent of CNA degradation after 1 h (Appendix A) or 2 h of exposure.

### 2.6. Role of Bacterial Motility in CNA Adaptation

Table 2 shows that 5/6 of the mutants with CNA MICs ≥ 1000 µg/mL showed mutations targeting the flagellar (*flgJ, flgG* or *fliE*) or pili (*pilJ*) machinery. Swarming assays that explore the functionality of both revealed a partial (CNA-B4 and -B6, that produce a Q_341_R variant of FlgG) or a complete loss of this type of motility (CNA-B3, -B5, and -B7, which contain a disrupted *fliE* gene). In contrast, CNA-B2 was able to fully swarm despite a P_83_L variation in protein FlgG (Figure 2A). As *P. aeruginosa* can attach to inert surfaces via its flagellum [23], its pili [24] or Pel polysaccharides [25], we investigated the capacity of the mutants to form biofilms in microplate wells and found the *fliE*-negative clones CNA-B3 to -B7 significantly impaired (Figure 2B). However, a possible link between the process of biofilm formation and CNA resistance was not confirmed with CNA-B1 (MIC = 1000 µg/mL), which behaved like the control strain to form a biofilm. To assess whether non-motile bacteria somewhat better survive CNA exposure, we constructed a *fliE*-negative PA14 mutant (PA14∆*fliE*) and compared its susceptibility to CNA with that of PA14, and a gain-of-efflux derivative PA14*mexAB^+^*. In support of our assumption, PA14∆*fliE* resisted CNA slightly better than PA14, but this resistance was lower than those conferred by NalC-dependent overproduction of MexAB-OprM (Figure 3). Finally, suggesting that some secondary mutations might somehow compensate for the energy cost potentially incurred by the increased efflux activity of *nalC* mutants, mutations of unknown significance were identified in genes *atpA* and *atpE*, which code for subunits of ATPase complex. Alternatively, this complex might play a role by itself in the resistance of *P. aeruginosa* to CNA as mutant A3 (MIC = 800 µg/mL) displayed a mutation in the ATPase β subunit but no MexAB-OprM dysregulation.

## 3. Discussion

Compared to many other Gram-negative species, *P. aeruginosa* naturally tolerates relatively high concentrations of CNA. This remarkable intrinsic resistance at least relies on CNA-induced expression of several RND efflux systems, and on still partially identified catabolic pathways that first convert the biocide into cinnamic alcohol [13]. In contrast to previous studies [15], the present work shows that a slightly higher resistance (from 1.3 to 1.6-fold) can be gained by *P. aeruginosa* toward this very toxic agent after several days of exposure. Most of the resistant mutants selected for further investigations overproduced the MexAB-OprM pump as a result of gene *nalC* alteration and exhibited cross-resistance to various antibiotics that are exported by this efflux system. These results confirm the key role played by MexAB-OprM in the protection of *P. aeruginosa* against a variety of natural antimicrobials present in EOs [12,26,27]. Lines of evidence support the notion that the phenolic moiety of some of these compounds is a structural motif recognized both by repressor NalC, and multidrug RND transporter MexB [13,28,29]. Inheritable alteration of NalC might also be the preferential way by which the pump is upregulated under environmental stress conditions, as this repressor was found to be systematically altered in *mexAB+* mutants isolated from soil [30], while those isolated in a clinical context exhibit alterations in NalC, MexR, or NalD [31]. As a plausible explanation to why the development of CNA resistance seems to be limited to low levels, we previously reported that CNA strongly induces MexAB-OprM expression for a short period of time until the biocide is degraded into the less toxic cinnamic alcohol, which is no longer an inducer for the pump [13]. The present paper demonstrates that the expression of *mexAB-oprM* in *nalC* mutants is at most 2.4-fold superior to the levels of wild-type cells exposed to CNA (as shown in Table 1; PA14+CNA). As for the *nalC* mutants emerging during our selection experiments, CNA-induced upregulation of MexAB-OprM in wild-type bacteria resulted in increased antibiotic resistance. If the constitutive derepression of *mexAB-oprM* in *nalC* mutants does not bring much in terms of CNA resistance, one could wonder why such mutants tend to emerge upon long term exposure to this agent. Our hypothesis is that when stably overproduced, MexAB-OprM better protects the whole population of a *nalC* mutant against fluctuating concentrations of CNA (especially near the minimum killing concentration) than when heterogeneously induced in individual wild-type cells.

In addition to *nalC* alteration*,* other mutations in a variety of genes tended to accumulate in the CNA-resistant clones of PA14, especially in those deficient in DNA repair system MutS. Though it could be tempting to make a link between all these alterations and the pleiotropic effects of CNA on bacterial cells, our results show that their contribution to the adaptation of *P. aeruginosa* to the electrophilic compound is likely to be weak, as they were not associated with a significant residual resistance when genes *mexAB* were deleted. Supporting the notion of stochastic alterations emerging in a context of hypermutator phenotype, most of them were related to unique genetic events targeting multiple cellular functions or structures, with no apparent link to CNA resistance. For instance, to assess the role of *pcaK*, a gene that encodes an aromatic acid H^+^ symporter [32] and whose activity is impaired in mutant CNA-B2, a deletion mutant from strain PA14 was constructed by allelic exchange. However, this construct showed the same resistance to CNA as PA14 (data not shown). Whether some secondary mutations could be involved in the maintenance of bacterial fitness remains unclear. Because the flagellar activity requires energy from the TCA cycle, its impairment by mutations might confer a fitness advantage when more energy is needed to degrade or export CNA. Recently, other researchers observed that ∆*flgG* or ∆*fliC* mutants of *P. aeruginosa* grow better than parental strain PA14 in presence of the glycopeptide antibiotic vancomycin [33]. Likewise, ∆*flgE* and ∆*flgD* mutants were found to have a better tolerance to gentamicin and colistin than their parent strain PAO1, when grown in biofilm [34]. Our experiments with a PA14∆*fliE* mutant failed to demonstrate a gain in gentamicin or colistin resistance, with respect to wild-type PA14 (data not shown). Thus, the role of FliE alteration Q54* in mutants CNA-B3, CNA-B5 and CNA-B7 remains questionable.

A collateral consequence of adaptation of *P. aeruginosa* to CNA, either through the induction or constitutive overexpression of the MexAB-OprM pump, is the decreased susceptibility of the pathogen to a number of antibiotics that are still widely used to treat patients. Though, to our knowledge, no clinical observations have been reported yet on the emergence of antibiotic resistant mutants in patients using EOs, the usefulness of these natural extracts in combination with antibiotics definitely needs to be evaluated as well as their effective concentration in the lung. In particular, the demonstration of an impact of electrophilic drugs such as CNA on evolution of hypermutator subpopulations residing in the CF airways would be of great clinical concern.

## 4. Material and Methods

### 4.1. Bacterial Strains, Plasmids and Growth Conditions

The reference strains, mutants and plasmids used in this study are listed in Appendix A. All the bacterial cultures were incubated at 37 °C in Mueller-Hinton broth (MHB) with adjusted concentrations of Ca^2+^ (from 20 to 25 µg/mL) and Mg^2+^ (from 10 to 12.5 µg/mL) (Becton Dickinson and Company, Cockeysville, MD, USA), or on Mueller-Hinton agar (MHA) (Bio-Rad, Paris, France) supplemented with antibiotics when required. Cinnamaldehyde (CNA) and dimethylsulfoxyde (DMSO) were obtained from Sigma-Aldrich (Saint-Quentin Fallavier, France).

### 4.2. Mutagenesis Frequency Evaluation

The mutagenic effect of CNA was determined by calculating the rate of mutants resistant to streptomycin as a result of point mutations occurring in the *rpsL* gene [19]. An overnight culture of *P. aeruginosa* PA14 was diluted into 20 mL MHB to reach A_600_ = 0.2 and incubated at 37 °C and 220 rpm for 3 h with a subinhibitory concentration of CNA (500 µg/mL); a similar assay was carried out in parallel with ciprofloxacin at 0.5 CMI (=0.006 µg/mL) as a positive control. Then, 50 µL volumes of the cultures were spread with a spiral plater (EasySpiral Pro, Interscience, Saint-Nom-la-Bretèche, France) on both MHA plates, and MHA plates supplemented with 8-fold the streptomycin MIC (final concentration: 512 µg/mL) (Sigma-Aldrich, Saint-Quentin Fallavier, France). The plates were incubated at 37 °C for 48 h, and the colonies were counted. Mutant rates were calculated by dividing the number of CFU developing on the streptomycin plates (mutation events) by the number of CFU on antibiotic-free plates [35]. The values presented are means of 3 independent experiments.

### 4.3. Selection of PA14 CNA Adaptation Mutants

An overnight liquid culture of *P. aeruginosa* strain PA14 was diluted to A_600_ = 0.8 in 20 mL of MHB containing 700 µg/mL CNA and incubated for 24 h at 37 °C under shaking (250 rpm). During ten consecutive days, 100 µL volumes were removed daily and spread on selective plates containing 900 µg/mL CNA. The remaining bacteria were pelleted by centrifugation (6400 *g*, 15 min), and resuspended in fresh MHB containing the same CNA concentration (Method A). Though 900 µg/mL CNA is toxic at the beginning of each transfer, *P. aeruginosa* is able to rapidly (in 3–6 h) metabolize this compound in less toxic molecules allowing surviving cells to grow for the 18 remaining hours. In another series of experiments (Method B), PA14 was submitted to an increased concentration of CNA (+50 µg/mL everyday) that reached 1400 µg/mL after 10 days.

CNA adaptation mutants were also obtained from three strains isolated from a CF patient (P6-1, P6-14 and P6-21 with CMI of CNA 300, 700 and 1000 µg/mL, respectively; Appendix A) after 18 h of exposure to toxic concentrations of CNA (400, 800, 1100 µg/mL, respectively).

### 4.4. Drug Susceptibility Testing

Minimum inhibitory concentrations (MICs) of selected antibiotics were determined by the standard serial two-fold dilution method, as recommended by the CLSI [36]. Experiments were carried out in 96-well microplates using the Freedom EVO^®^ robotic system (TECAN, Männendorf, Switzerland) with inocula of 5.10^5^ CFU/mL. MIC values of CNA (DMSO 1%) were determined in MHA with inocula of 10^6^ CFU per spot. In both cases, bacterial growth was visually assessed after 18 h of incubation at 37 °C. To confirm the results, MIC experiments were performed in triplicate. Drug susceptibility of CNA-resistant mutants deriving from CF isolates P6-1, P6-14 and P6-21 was assessed by the agar diffusion method with Bio-Rad disks and MHA, according to current EUCAST protocol (https://www.eucast.org/ast_of_bacteria/, accessed on 1 September 2022).

### 4.5. RT-qPCR Experiments

Specific gene expression levels were measured by quantitative PCR after reverse transcription (RT-qPCR), as previously described [37]. RNA was reverse transcribed with ImProm-II reverse transcriptase as specified by the manufacturer (Promega, Madison, WI, USA). Amounts of specific cDNA were quantified on a Rotor Gene RG6000 instrument (Qiagen, Courtaboeuf, France) by using the QuantiFast SYBR green PCR Kit (Qiagen), and primers annealing to the target genes (Appendix A). For each strain, mRNA levels of target genes were normalized to that of housekeeping gene *rpsL* and were expressed as a ratio to the transcript levels of strain PA14. Mean gene expression values were calculated from two independent bacterial cultures, each assayed in duplicate. As shown previously, transcript levels of *mexB* > 2-fold, *mexY* ≥ 5-fold, *mexC* and *mexE* ≥ 20-fold those of PA14, were considered as significantly increased because associated with a ≥two-fold higher resistance to respective pump substrates [37,38].

### 4.6. Construction of PA14 fliE-Defective Mutant

In order to study the role of motility in bacterial tolerance to CNA, a defective mutant in *fliE* was constructed from wild-type strain PA14. Gene inactivation was carried out using the suicide plasmid pKNG101 (Appendix A) and homologous recombination events. Briefly, recombinant plasmids were constructed by assembly cloning using the NEBuilder Hi-Fi DNA Assembly Cloning Kit (New England Biolabs, Evry, France), and the deleted chromosomic region amplified with appropriate primers (Appendix A). Assembly products were directly used to transform competent *Escherichia coli* strain CC118λpir (Appendix A). Recombinant plasmids with appropriate inserts were transferred to *P. aeruginosa* PA14 by conjugation. Transconjugants were selected on Pseudomonas Isolation Agar (PIA; Becton Dickinson) containing 2000 μg/mL streptomycin. Excision of the undesired pKNG101 sequence was obtained by plating transformants on M9 plates (8.54 mM NaCl, 25.18 mM NaH_2_PO_4_, 18.68 mM NH_4_Cl, 22 mM KH_2_PO_4_, 2 mM MgSO_4_, 0.8% agar, pH 7.4) containing 5% (wt/vol) sucrose. Negative selection on streptomycin-containing MHA allowed the identification of transconjugants that had lost the plasmid. Finally, the allelic exchange was checked by PCR and confirmed deletion of 198-bp in gene *fliE.*

### 4.7. Genome Sequencing and in Silico Analysis

The whole DNA content of mutants CNA-A1 to A3 and CNA-B1 to -B7 was sequenced with the Illumina technology. Briefly, total bacterial DNA was extracted from overnight cultures by using the PureLink Genomic DNA Mini Kit (ThermoFisher Scientific, Illkirch-Graffenstaden, France), then was quantified with a NanoDrop^®^One spectrophotometer (ThermoScientific, Villebon-sur-Yvette, France), and finally sequenced by Microsynth AG (Balgach, Switzerland) on an Illumina NextSeq sequencer with v2 chemistry, using 2 × 150 paired-end reads. DNA libraries were prepared with Nextera XT DNA Library Preparation Kit (Illumina, San Diego, CA, USA). High throughput sequencing yielded from 3,355,318 to 5,529,638 paired reads per mutant (6,537,648 for PA14), with a coverage depth comprised between 150 and 250 X. The reads were trimmed with Sickle [39] using a minimum length of 75 bp and a minimum base quality of 20, subsampled to 80 X by home-made script, and then aligned using BWA-MEM [40] against the PA14 reference genome (GCF_000014625.1), using default parameters. Variant calling was performed with Freebayes [41] in haploid mode, a minimum quality of base call and mapping of 20, a minimum alternate fraction of 0.8, and a minimum alternate count of 10. Sequence variations, found in the mutants and in the wild type PA14 strain, were removed.

Phylogenetic tree was built with MrBayes using the concatenation of identified SNPs from the ten mutants CNA-R. The analysis was performed with the HKY85 mutation model, and a gamma-distributed rate variation [42] during 1 million iterations.

Complete genomic sequences of the *P. aeruginosa* mutants CNA-A1 to CNA-A3, and CNA-B1 to CNA-B7 mutants have been deposited in the NCBI data base under the BioProject accession number PRJNA663565. Mutations found in genes of mutants CNA-B1 to -B7 are listed in Appendix A. SIFT (https://sift.bii.a-star.edu.sg/, accessed on 1 September 2022) and PROVEAN (cutoff −2.5) (http://provean.jcvi.org/index.php, accessed on 1 September 2022) algorithms were used to predict the effects of identified substitutions. Three-dimensional structural model of NalC (UniProt reference Q9HXS0) was obtained by homology modeling using MODELLER software [43], with 3BHQ as a template structure. Sequence alignment was obtained with ENDscript [44], and figures were generated with PyMOL Molecular Graphics System, Version 2.0 Schrödinger, LLC, New York, NY, USA.

### 4.8. Motility and Cell Adhesion Assay

#### 4.8.1. Swarming

Swarming motility is due to the combined action of rhamnolipids, type IV pili and flagella, and was evaluated on a semi-solid surface. Positive swarming strains form macroscopic colony patterns characterized by the coordinate translocation of bacteria on the semi-solid medium, when grown under optimal conditions. Briefly, 5 µL of a bacterial suspension (0.2 McFarland) were spotted onto the surface of an M8-Swarm medium (Minimal medium. NaCl 8 mM, NaH_2_PO_4_ 42 mM, KH_2_PO_4_ 22 mM pH 7.4 supplemented with glucose 0.2%, MgSO_4_ 2 mM, Casaminoacids 0.5%, and agar 0.5%). Swarming colonies were observed after an incubation of 24 h at 37 °C.

#### 4.8.2. Detection of Biofilm Formation by Adherence Test

200 µL fractions of bacterial suspension (10^8^ CFU/mL) were incubated 24 h at 30 °C into a polystyrene 96-well microplate. Each well was washed twice using 200 µL of distilled water to eliminate planktonic bacteria. Biofilm was colored using 200 µL of crystal-violet solution at 1% (*w*/*v*). After 15 min at room temperature, the wells were washed twice using 200 µL of distilled water, and crystal-violet attached to sessile bacteria was solubilized with 300 µL ethanol at 99% (*v*/*v*). Finally, the absorbance was measured at 600 nm, against a negative control containing 200 µL of cMHB [45].

### 4.9. Metabolite Extraction and Detection Using Thin Layer Chromatography (TLC)

Standard working solutions of CNA (cinnamaldehyde) and its metabolite CN-OH (cinnamic alcohol) were prepared by diluting aliquots of >98% stock solutions in DMSO. Overnight bacterial cultures were diluted into 20 mL of fresh MHB, and incubated with shaking (250 rpm) at 37 °C. When the cultures reached an absorbance of *A*_600 nm_ = 0.8, CNA was added to a final concentration of 1 mg/mL. After one and two hours of exposure, the growth medium was collected by centrifugation and filtration through two filters of 0.45 and 0.2 µm pore size, successively. An organic extraction was repeated three times using 25 mL dichloromethane (for a total volume of 75 mL). The organic fractions were pooled and dried under rotative evaporator, and finally redissolved in 1 mL methanol. Standards were diluted (1:100) in methanol. A fraction of 25 µL for sample and 2 µL for standards was sprayed as 8 mm bands on a TLC plate (Alugram^®^ Xtra SIL G UV254, Macherey-Nagel, Hoerdt, France) using an automatic sampler (ATS4, Camag, Moirans, France) connected to visionCATS Camag TLC software V2.4. The TLC plate was developed in an automatic developing chamber (ADC 2, Camag) with a mobile phase containing cyclohexane:ethyl acetate (7:2) over a 70 mm developing distance. Spots were observed using UV-light at 254 nm (CV-415.LS, Uvitech, Cambridge, UK).

## Figures and Tables

**Figure 1 antibiotics-11-01790-f001:**
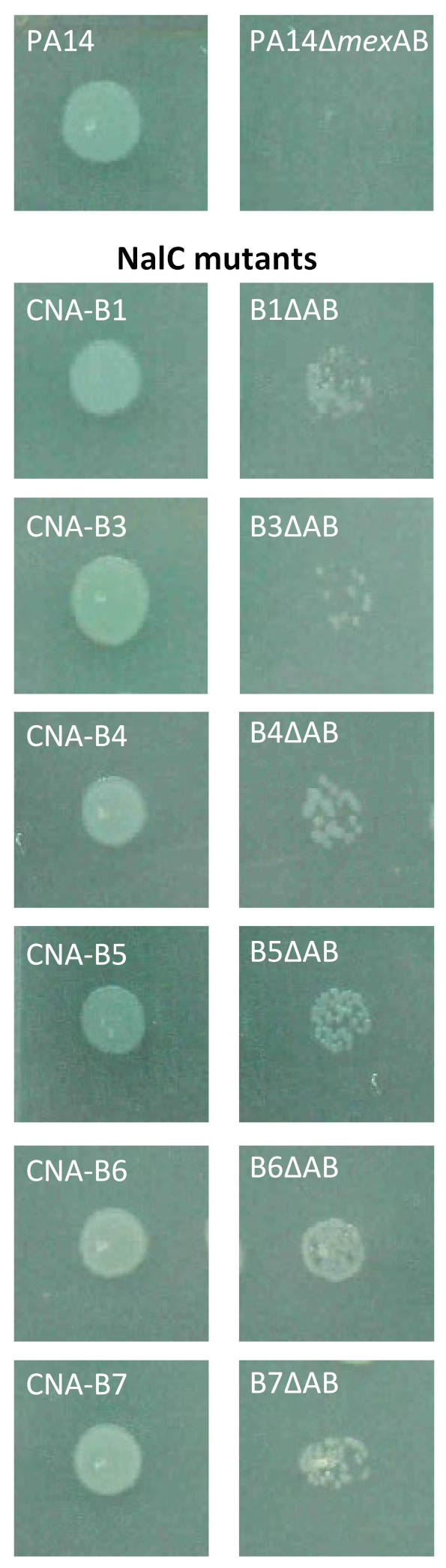
Impact of genes *mexAB* deletion on resistance to CNA. Genes *mexAB* were deleted in PA14 (CNA MIC before deletion equal to 700 µg/mL), and CNA-resistant *nalC* mutants CNA-B1, B3, and B5 (1000 µg/mL), and CNA-B4, B6, B7 (1100 µg/mL). The inactivated mutants (∆AB) and their progenitors were spotted onto the surface of an MHA plate containing 500 µg/mL CNA.

**Figure 2 antibiotics-11-01790-f002:**
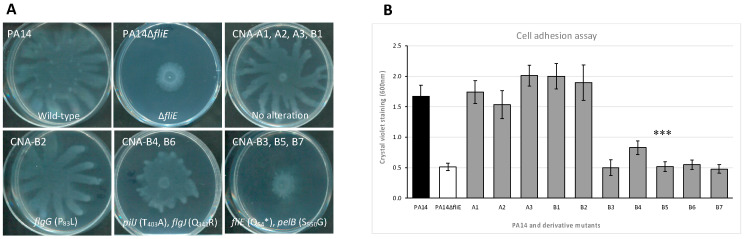
Motility and initial surface attachment of CNA-resistant mutants. (**A**) Swarming motility: the mutations found in genes responsible for flagellar structure/assembly, pili formation, and biosynthesis of biofilm matrix are indicated. (**B**) Absorbance at 600 nm is representative of sessile bacteria stained by crystal-violet in microplates (mean values resulting from 10 replicates; Student’s t-test, *** *p*-value < 0.001).

**Figure 3 antibiotics-11-01790-f003:**
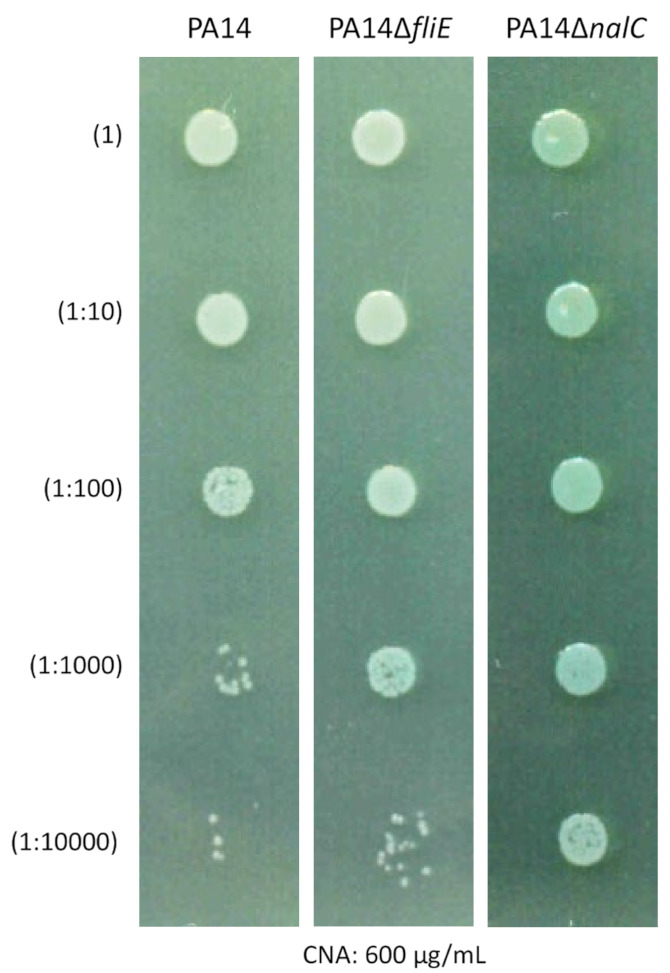
Increased resistance of PA14∆*fliE* to CNA. Serial Log_10_ dilutions of strain PA14 and its derivative mutants PA14∆*fliE* and PA14∆*nalC* all calibrated at 0.5 McFarland were spotted onto the surface of an MHA plate containing 600 µg/mL CNA.

**Table 1 antibiotics-11-01790-t001:** Features of CNA resistant mutants.

Strains/Mutants ^a^	Sequence	Transcript Levels ^d^					MICs (µg/mL) of Antibiotics and CNA ^e^						
	*nalC*	*mexB*	*armR*	*mexY*	*mexC*	*mexE*	TIC	AZT	CIP	IPM	CST	GEN	CNA
PA14	WT	1	1	1	1	1	32	4	0.06	0.5	1	0.5	700
PA14 + CNA ^b^	WT	**3.2**	**43**	5.5	118	100	64	16	1	0.5	2	2	-
PA14*mex*AB^+ c^	WT	**6.5**	1	ND	ND	ND	128	16	0.25	0.5	1	0.5	900
PA14∆AB	WT	-	ND	ND	ND	ND	16	≤0.5	0.06	0.5	1	0.5	500
CNA-A1	**−A_486_**	**3.4**	**61**	0.2	0.5	1.3	128	32	0.25	0.5	0.5	0.125	900
CNA-A2	**T_24_P**	**2.5**	**24**	0.1	0.1	0.2	256	16	0.25	0.5	0.5	0.25	900
CNA-A3	WT	1.3	1	0.5	0.7	0.4	64	8	0.06	0.5	0.25	0.125	800
CNA-B1	**+C_450_**	**4.9**	**164**	0.3	0.4	0.8	>256	64	0.5	0.5	0.5	0.25	1000
CNA-B2	WT	1.2	1	0.5	0.6	0.5	64	16	0.06	0.5	0.5	0.25	800
CNA-B3	**+C_450_**	**5.7**	**146**	1.1	0.4	0.9	>256	64	1	1	4	1	1000
CNA-B4	**+C_450_**	**3.3**	**130**	0.4	0.5	0.9	>256	64	0.5	2	1	0.125	1100
CNA-B5	**+C_450_**	**6.0**	**208**	1.1	0.7	0.9	>256	64	0.5	2	4	1	1000
CNA-B6	**+C_450_**	**5.0**	**136**	0.7	0.6	0.8	>256	64	0.5	1	1	0.5	1100
CNA-B7	**+C_450_**	**7.6**	**423**	2.5	0.6	1.6	>256	64	1	2	4	1	1100

^a^ Mutants A1-A3 were obtained after 10 days of exposure to a fixed 900 µg/mL CNA concentration; mutants B1-B7 were isolated after exposure to increasing concentrations of CNA: up to 1100 µg/mL (B1-B3) and up to 1400 µg/mL (B4-B7). Genetic data of the eight MexAB-OprM-overproducing mutants derived from parent strain PA14 are indicated in boldface. ^b^ NalC derepression resulting from 30 min exposure to 512 µg/mL CNA (results published in [13]). ^c^ In this control strain, overproduction of MexAB-OprM is due to a 3-bp deletion in repressor gene mexR (nalB mutant). ^d^ Expressed as a ratio to wild-type reference strain PA14. Mean values were calculated from two independent bacterial cultures each assayed in duplicate. ^e^ TIC: ticarcillin; CIP: ciprofloxacin; IPM: imipenem; CST: colistin; GEN: gentamicin; CNA: cinnamaldehyde.

**Table 2 antibiotics-11-01790-t002:** Main genetic alterations of the CNA resistant mutants.

Proteins	A-Type Mutants (CNA MIC in µg/mL)			B-Type Mutants (CNA MIC in µg/mL)							Protein Functions
	CNA-A1 (900)	CNA-A2 (900)	CNA-A3 (800)	CNA-B1 (1000)	CNA-B2 (800)	CNA-B3 (1000)	CNA-B4 (1100)	CNA-B5 (1000)	CNA-B6 (1100)	CNA-B7 (1100)	
											**DNA mismatch repair**
**MutS**				**−9 pb**	Q_589_R	**−9 pb**	**−9 pb**	**−9 pb**	**−9 pb**	**−9 pb**	DNA mismatch repair protein MutS
											**Transmembrane transport**
**NalC**	**-A_486_**	**T_24_P**		**+C_450_**		**+GG_442_**	**+C_450_**	**+C_450_**	**+C_450_**	**+C_450_**	Transcriptional repressor NalC of MexAB-OprM pump
**DppB**				**G_114_D**		**G_114_D**	**G_114_D**	**G_114_D**	**G_114_D**	**G_114_D**	Dipeptide ABC transporter permease DppB
											**Energy production**
**AtpD**			**P_305_S**								F0F1 ATP synthase beta subunit
**AtpA**						D_290_G		D_290_G			F0F1 ATP synthase alpha subunit
**AtpE**							T_23_A				F0F1 ATP synthase C subunit
											**Response to oxidative stress**
**MxtR**							**W_1071_***		**W_1071_***	L_1068_P	Orphan sensor kinase MxtR
**YbbN**						**F_86_S**	**F_86_S**	**F_86_S**	**F_86_S**	**F_86_S**	Thioredoxin (oxidoreductase)
											**Response to chemicals**
**PcaB**						E_239_G		E_239_G		E_239_G	3-carboxy-cis, cis-muconate cycloisomerase
**TlpQ**				**L_561_P**		**L_561_P**	**L_561_P**	**L_561_P**	**L_561_P**	**L_561_P**	Chemotaxis transducer TlpQ
											**Motility**
**FliE**						**Q_54_***		**Q_54_***		**Q_54_***	Flagellar hook-basal body protein FliE
**FlgJ**							Q_341_R		Q_341_R		Flagellar rod assembly protein FlgJ
**FlgG**					P_83_L						Flagellar basal body rod protein FlgG
**PilJ**							**T_403_A**		**T_403_A**		Twitching motility protein
											Nucleotide–sugar metabolic process
**RbsR**						V_320_M	V_320_M	V_320_M	V_320_M	V_320_M	Ribose operon repressor RbsR
											** *Unknown functions* **
**PA0841**						A_304_V		A_304_V			Hypothetical protein
**PA3283**						N_142_S		N_142_S		N_142_S	Hypothetical protein

* STOP codon. Mutations predicted as being deleterious to protein function by PROVEAN (score < −2.5) are indicated in boldface.

## Data Availability

Raw data are available from authors at request.

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
