# Peer review of "Exposure of Pseudomonas aeruginosa to Cinnamaldehyde Selects Multidrug Resistant Mutants"

_antibiotics, 2022, doi:10.3390/antibiotics11121790_

Round 1

Reviewer 1 Report

A very interesting article. The research is very well documented. The authors showed that  exposure of wild-type strain Pseudomonas aeruginosa (PA14)  for 5 to 10 days to fixed (900 μg/mL) or increasing (from 900 to 1,400 μg/mL) concentrations of the cinnamaldehyde (the main component of cinnamon essential oil) resulted in emergence of resistant mutants. Eight mutants (from 10 selected) showed increased resistance to antibiotic molecules exported by the pump including ß-lactams and fluoroquinolones. Authors speculated that the usefulness of this natural extracts in combination with antibiotics definitely needs to be evaluated. My question is: Is the dose of cinnamaldehyde used to treat mild infections of Pseudomonas aeruginosa high enough to lead to mutants of these bacteria.

Author Response

Answer: we thank the reviewer 1 for this relevant comment.

To our knowledge, there is no scientific report about the use of essential oils by CF patients nor the dose consumed. At Besançon, a local survey showed that 30% of CF patients use these natural products, but it is difficult to know exactly the quantities since their use is not regulated. There is also no information about the diffusion of these volatile compounds and their effective concentration in the lungs. In order to improve knowledge on this subject, the French national association vaincre la mucoviscidose (https://www.vaincrelamuco.org/) has just put a questionnaire online about the use of essential oils by those patients. In our lab, we are currently testing the efficacy of some natural extracts directly on the sputums of CF patients to measure the impact on bacterial populations.

We added a comment in the discussion section (Lines 307-308).

Reviewer 2 Report

Tetard et al. challenged Pseudomonas aeruginosa to cinnamaldehyde (CNA). Three CNA resistant strains—A1–A3—were obtained under the constant CNA pressure—900 µg/mL, which is slightly higher than the MIC (700 µg/mL). Seven CNA resistant strains—B1–B7—were obtained under an increasing CNA pressure series—from 900 to 1,400 µg/mL. They sequenced the genomes of the mutant strains. They found a single SNP in each of A1–A3, and multiple SNPs in each of B1–B7. Eight of the 10 strains contained the mutations in nalC (regulating the efflux pump MexAB-OprM). Of the six mutant strains with the highest CNA resistance levels, five lost motility because of mutations in flgJ, fliE and/or pilJ. The science is solid. The manuscript is well-written. 

My only comment is that the lack of mutations observed could be due to the passaging setup. The population size is 20 mL. During each transfer, a volume of 19.9 mL (= 20 mL - 100 µL) were pelleted and then resuspended in fresh broth. I am not sure if this allowed much room for growth. I am guessing more mutations would show up if under a chemostat setup. I am listing some good papers applying chemostats—chemostat (doi.org/10.1111/1574-6976.12082), turbidostat (doi.org/10.1093/gbe/evz197), morbidostat (doi.org/10.1038/ng.1034).

Minor comments to avoid confusion:

1. Lines 15–16, "CNA-A1 to A3 differed from PA14 by single SNPs" → "each of CNA-A1 to A3 differed from PA14 by one SNP". 

2. Line 18, "and harbored multiple SNPs" → "each of them harbored multiple SNPs".

Author Response

Answer: We thank the reviewer 2 for this comment.

Between each transfer, cells are growing at 37°C for 24h under agitation, that should allow P. aeruginosa cells to replicate several times. Indeed, even if the concentration of CNA is toxic at the beginning of each transfer, P. aeruginosa rapidly (in 3-6 hours) metabolizes this compound in less toxic molecules allowing surviving cells to grow for the 18 remaining hours.

We added a comment in the materials and methods section (Lines 339-341).

Minor comments to avoid confusion:

  1. Lines 15–16, "CNA-A1 to A3 differed from PA14 by single SNPs" → "each of CNA-A1 to A3 differed from PA14 by one SNP". 

Answer: it has been corrected lines 15-16.

  1. Line 18, "and harbored multiple SNPs" → "each of them harbored multiple SNPs".

Answer: it has been corrected line 18.
